mechanical engineering/biomedical engineering/materials science

frugal manufacturing, sustainability, climate change, net finished shape, pandemic

**Author for correspondence:**
Balkrishna C. Rao
e-mail: balkrish@iitm.ac.in

# Frugal manufacturing in smart factories for widespread sustainable development

## Balkrishna C. Rao

Department of Engineering Design, Indian Institute of Technology Madras, Chennai 600036, Tamil Nadu, India

(iD) BCR, 0000-0002-5984-8538

Manufacturing is a crucial activity of product development that feeds into and is also influenced by the design process. Any material conservation gained during manufacturing directly affects the green credentials of a product. Manufacturing waste can be contrived to approach zero through a recently developed *frugal design approach* that quantifies resource conservation at all stages of development of a product engineered for frugality. Accordingly, this effort presents *frugal manufacturing* (FM), integral to the *frugal design approach*, for utmost reduction of waste while aiming for good surface integrity, better properties, minimal number of processes and low cost. Other than saving on energy and hence emissions, the new concept of FM also goes beyond current *near net shape* technologies, which advocate mainly for zero wastage and suitable properties while using a narrow range of manufacturing processes. Case studies involving *high-speed machining*, *superplastic forming* and *additive manufacturing* of aerospace alloys have been presented that bring out the features and benefits of FM. As such the multipronged objectives of FM should be dovetailed with those of *smart factories* for creating novel technologies that abet widespread sustainable development. Such enhancement of the *smart factories* concept has been argued to support unusual applications such as the fight against pandemics including the current one involving COVID-19.

## 1. Introduction

Frugality in resource consumption is an important pillar of *sustainable development* [1,2]. This has been exemplified, in recent years, by the proliferation of both grassroots and sophisticated or *advanced frugal* products, which provide good (or even better) functionality, vis-à-vis existing conventional types, at low cost

while consuming minimal resources [3–8]. Examples of low-cost electrocardiograms [1], portable ultrasound [1], lighting without power source [7] and cars [1], among other examples [1,7], attest to the potential of *frugal* products for widespread *sustainable development*. The *frugal* nature of these products fits in well with the cradle-to-cradle concept of sustainability promoted by the *Environmental Protection Agency* (EPA) [9] and the *National Academy of Engineering* [10,11] to make manufacturing strong and competitive [12]. Although systematic design [13,14] and subsequent complex networking [15] of products together with suitable policymaking and educational programmes for their diffusion [16] are crucial, manufacturing aspects are equally important for their successful realization. This is because fabrication of these products needs judicious use of manufacturing and/or assembly for lowering consumption of raw materials. Moreover, manufacturing itself plays a significant role in providing crucial inputs to the *frugal design approach* [13]. Therefore, there is a need to make manufacturing *frugal*.

It should be noted that this effort differentiates between *near net shapes* (NNSs) [17] and *net finished shapes* (NFSs) with the former needing some form of finishing operation(s) to arrive at an NFS. In other words, an NFS is the final product ready for application.

# 2. Frugal manufacturing

This effort defines *frugal manufacturing* (FM) as fabrication using a minimum number of low-cost processes producing zero waste for creating an NFS possessing necessary geometrical tolerances; requisite *surface integrity*; and appropriate properties. *Surface integrity* refers to the finish and accompanying sub-surface residual stresses of the product and properties refers to any additional requirements such as mechanical ones of strength, toughness, etc or any characteristic required for functionality. In other words, NFSs of requisite quality are to be produced at low-cost with ideally a single process using enough quantity of material for avoiding any extra operations, including finishing and, their resulting waste. In fact, cost reduction will be further buttressed through both avoidance of waste due to concomitant need for less stock of feed-material and a single process that negates the outlays over multiple ones. The restraint on number of processes would also lower power consumption and the attendant emissions of *greenhouse gases* (GHGs).

Overall, FM subsumes the definition for an ideal NNS given by Altan & Miller [18]. Some efforts on NNS document both reduction of waste in specific manufacturing processes and achievement of specific properties with low defect rates. This is borne out by few existing papers [17,19,20] that report aiming for a *near net shape* while also focusing on attendant functionality, mechanical properties and microstructure without explicit reference to *surface integrity*. *Surface integrity* considerations are significant because of their potential to enhance product life [21,22], which is vital to *sustainable development*. Also, NNS processes are typically *bulk* and *sheet* deformation operations [23] involving large plastic strains, whereas FM includes the entire gamut of additive and subtractive processes. Therefore, FM is a general concept that also aims for these additions not available in NNS processes: proper *surface integrity* of the product; consideration of a wider range of processes; minimal number of processes; and low-cost process(es).

The increasing numbers of *advanced frugal* products [7] make it imperative to use minimal manufacturing for creating NFSs, including intricate geometries, without secondary finishing operations such as machining and grinding. This is because manufacturing is a critical input to the *frugal design* of such products and hence needs to generate as little waste as possible to qualify for *frugality* [13,14]. In particular, ideally, a *net finished shape* with zero wastage would help leading to maximum improvement in the *factor of frugality* of a product [13,14]. It is noteworthy that even without a *frugal* mindset, the premise underlying FM will go a long way in supporting *sustainability* through less wastage; lower costs due to cost-effective processes and also attendant savings in materials wasted; lower power consumption; and low-cost quality products. In fact, savings in resources are especially important when processing expensive materials like superalloys [17].

## 2.1. Why frugality in smart factories?

The potential of FM to achieve high quality, zero waste and low cost in a minimal number of operations should be harnessed for the development of *smart factories* which are vital to Industry 4.0 [24]. The lesser numbers of operations—ideally a single operation—also makes FM an energy saver, which is critical to Industry 4.0. Inclusion of FM in Industry 4.0 is essential since FM controls actual operations based on their relevant physics, which Industry 4.0 does not. But realization of FM in real-time *smart factories*

will require, *inter alia*, interactions between machine-tool systems and existing tool-kit of Industry 4.0 including digital twins and *artificial intelligence* (AI), to name a few. In fact, building of novel machine tools, designed from the perspective of FM, would be greatly aided by these pillars of Industry 4.0. All in all, incorporating FM into Industry 4.0 will make factories truly 'smart' by 'frugalizing' their process(es) against climate change.

# 3. Candidate processes for frugal manufacturing

This section covers primary ways of manufacturing for metals that qualify for FM. Any other real-time process could be viewed as a variant of these operations—some of which are also described in this section. Although these processes have been listed separately to show their individual potential for going *frugal*, their combinations, as explained later, also aid frugality. In other words, these processes are the current building blocks of FM with possible foreseeable revisions.

## 3.1. Casting

*Casting* is a popular *classical* manufacturing process [25] that involves production of NNS through pouring; subsequent filling of mould and resulting solidification of parts. Cast parts typically need post-treatments including heat and/or finish machining for imparting proper *surface integrity* and also necessary properties to the NNS. Variants to casting include NNS processes such as *pressure infiltration casting* [26] and *gel casting* of metallic powders [27] while *semi-solid casting* can create NFS [28].

## 3.2. Bulk deformation and sheet processing

These operations use severe plastic flow to arrive at parts both plain and intricate. Examples, to name a few, include extrusion, forming, rolling and forging for *bulk* deformation processes and blanking and shearing for *sheet metal* operations. The *bulk* and *sheet metal* processes involve minimal to no change in the volume of feed material used for fabrication. Since volumetric change is absent in plastic deformation, volume here refers only to that of the feed material. Moreover, production of intricate patterns and shapes is possible [29] through hot working and feed materials that show *superplastic* behaviour as in *semi-solid forming* [30,31]. The work by Tateno [25] attests to the capability of these processes in fabricating large NNSs. Since these processes hew very close to the finished part with minimal attendant finishing operations, they possess the ability to directly produce a *net finished shape* [23]. In fact, roll forming can directly create NFSs for even complex geometries [32]. However, since the quantum of feed stock going into any *bulk* or *sheet* operation is limited, the stock quality in terms of material defects, homogenity in composition, surface texture and other features should be superior for creating defect-free products [19].

## 3.3. Additive manufacturing

The *additive manufacturing* (AM) of metallic materials has come a long way to create products with complex shapes; repair worn parts and generate parts with composition-gradient [33]. In so doing, AM is not ridden with problems such as machine-tool chatter that plague conventional or subtractive processes [34]. AM or *3D printing* uses layer-by-layer deposition to build a part from the ground up. Some AM techniques such as *direct metal deposition* (DMD) can be looked upon as a variant of the *casting* process. The resulting parts hew closely to the required shape but typically possess inadequate surface finish, improper residual stresses and also possibly anisotropy in properties. Hence, AM processes typically produce NNSs [33] needing subsequent heat treatments and/or finishing operations.

## 3.4. Powder metallurgy

Metallic powders are moulded by *powder metallurgy* (PM) techniques into green compacts and subsequently sintered into NNS including intricate geometries. PM techniques are employed in several sectors to create products with requisite properties. However, porosity is an ever present problem [35] in PM products that hampers both bulk properties and *surface integrity*. A majority of PM techniques produce NNSs needing subsequent heat treatments, not least for minimizing porosity, and also finishing operations [36].

## 3.5. Non-traditional techniques

*Non-traditional manufacturing* (NTM) applies non-traditional media such as water for processing materials into products [37]. An example of NTM is *abrasive water jet* (AJW) cutting which uses a water-based abrasive slurry aimed under very high pressures to cut materials and hence make parts. These techniques typically remove very little material and hence are potential candidates for FM. However, NTM also results in NNSs that need post-treatments and secondary finishing operations for achieving suitable properties and proper *surface integrity*. Besides, NTM processes are also energy intensive.

## 3.6. Metal cutting

*Metal cutting* operations create parts by subtraction of material and are either of the machining or abrasive-cutting type. The former uses a cutter with well-defined geometry while the latter uses random grains, such as those of sand, bonded together in a wheel for cutting materials. In either type, excess material is removed to make a product against constraints of *surface integrity* and dimensional tolerances. *Metal cutting* typically follows the *frugal* processes of this section as a finishing operation. However, in recent years, *high-speed machining* (HSM) has shown the ability to manufacture features in a single pass, an example of which has been subsequently presented in this work as a case study. Moreover, *metal cutting* operations are also being optimized for hewing to product shapes and hence remove the least amount of stock for achieving requisite tolerances [38], all akin to production of NNSs. Also, recently, *metal cutting* has been unusually used as a single-pass *severe plastic deformation* (SPD) process for producing novel *ultra-fine-grained* (UFG) materials [39] whose lesser usage, due to their superior mechanical properties, create *frugal* products.

# 4. Gauging frugality of processes

FM's goal of restricting waste and maintaining both appropriate properties and good *surface integrity* while costing less can be realized by the individual processes outlined in the previous section or their combinations. In this regard, an existing scheme based on the *factor of frugality* can be used to quantify the streamlining of manufacturing processes going into the making of a product, *frugal* or otherwise.

## 4.1. Factor of frugality

The *factor of frugality* ($F$), underlying the *frugal design approach* (FDA), facilitates systematic design of *frugal* products [13,14]. The *factor of frugality* is an extension of the *classical* and popular *factor of safety* ($S$) to account for restraints on resource consumption, which is vital to *sustainable development*. It is the sum total of the *factor of safety* and *material saved* (MS) parameters, which account for savings in addition to that realizable from a low $S$ design. In other words, *frugality* is achieved in a systematic manner by starting with a low $S$, approximately 1.5, design and subsequently selecting extraneous materials- and design-related features to save yet more material for this design. Both $F$ and $S$ are symbolically summarized in $F^S$ with the defining equation for $F$ being

$$F = S + \sum_{i=1}^{n} MS_i. \tag{4.1}$$

Here $i$ indexes through various extraneous schemes for conserving material including simple design, biomimetics, materials, manufacturing and salvaging [13,14]. The $MS$ parameter associated with manufacturing, i.e. FM, is denoted by $MS_{MANU}$. In particular, $MS_{MANU}$ quantifies material wasted in manufacturing, e.g. as chips in *metal cutting* operations and, hence is a crucial parameter underlying FM. Accordingly, the defining equation for $MS_{MANU}$ is given by,

$$MS_{MANU} = \frac{(W_{MAX} - W_{MIN})}{W_{MAX}}, \tag{4.2}$$

where $W_{MAX}$ refers to actual manufacturing operation(s) maxing out on waste production while $W_{MIN}$ refers to operation(s) reducing waste to its lowest level and therefore selected for FM. Therefore, FM's aim of minimizing waste is achieved through maximization of $MS_{MANU}$. However, maximization of $MS_{MANU}$ is sought in this work against constraints on costs and product-quality in terms of

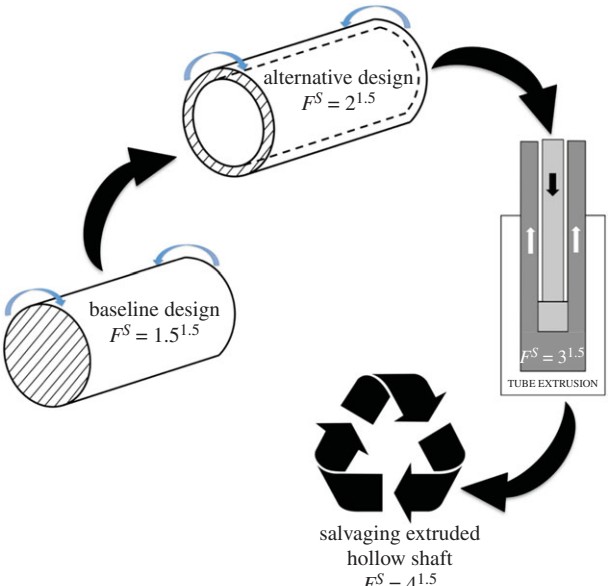

**Figure 1.** Improvement in *factor of frugality* of a shaft through various *material saving* schemes, including manufacturing, considered in a natural sequence [14].

**Table 1.** Computing factor of frugality for a shaft [14]. (material: steel; yield shear strength: 300 MPa; shear modulus: 80 GPa).

| | | material saved | | | |
|---|---|---|---|---|---|
| S. no. | factor of safety ($S$) | alternative design ($MS_1$) | manufacturing ($MS_2$) | salvaging ($MS_3$) | factor of frugality ($F$) |
| 1 | 1.5 | 0 | 0 | 0 | 1.5 |
| 2 | 1.5 | 0.5 | 0 | 0 | 2.0 |
| 3 | 1.5 | 0.5 | 1 | 0 | 3.0 |
| 4 | 1.5 | 0.5 | 1 | 1 | 4.0 |

tolerances, properties and *surface integrity*. In other words, $MS_{MANU}$ should ideally be unity by accounting for a single-pass process that is low-cost and imparts necessary properties and also *surface integrity* to the part. This would also account for the need to enshrine *surface integrity* requirements in the *factor of frugality* as expounded by Rao [13]. In other words, the model for $F^S$ should also stress on achieving the correct *surface integrity* in addition to properties by focusing on $MS_{MANU}$.

### 4.1.1. Example on factor of frugality

The computation of $F^S$ is exemplified by a *frugal shaft* from Rao [14]. The simple example of a shaft—a popular engineering component—is useful in understanding the quantification of relevant *MS* parameters and subsequently $F^S$. The shaft, made of steel, has to be designed for transmitting a power of 500 kW at 1200 r.p.m. under torsional loads. The details of the classical design process have been outlined in Ugural [40]. Accordingly, table 1 and figure 1 present the *frugal* design of the shaft where $MS_2$, i.e. manufacturing's share $MS_{MANU}$, is just one of the *material saved* parameters going into the computation of $F^S$.

The first step involves fixing the *factor of safety* at a low admissible value of 1.5 throughout quantification. A $S$ of 1.5 by itself creates a 'leaner' shaft that avoids excess material accruing with increasing $S$. The next step listed as first entry in table 1 concerns baseline design, i.e. classical design devoid of frugality, of a solid or 'bulky' shaft for the $S$ of 1.5. This results in a solid shaft of diameter 47 mm. The *MS* schemes are next applied with the second entry, on the *material saving* scheme of *alternative design*, listing material savings of 50% (or an $MS_1$ of 0.5) by designing a hollow shaft, which is the lightest candidate in the pool of possible designs for the shaft. The resulting outer and inner

**Table 2.** Ideal potential for stand-alone *frugal manufacturing* in individual processes.

| no. | manufacturing process | $MS_{MANU}$ | output |
|---|---|---|---|
| 1 | casting | 1 | net finished shape |
| 2 | additive manufacturing (AM) | 1 | net finished shape |
| 3 | bulk deformation and sheet processing | 1 | net finished shape |
| 4 | non-traditional manufacturing | 1 | net finished shape |

diameters of the hollow shaft are 56 and 44 mm, respectively. Subsequent steps/entries consider *MS* schemes on *manufacturing* and *salvaging* in this respective sequence. Manufacturing is accomplished by extrusion as an NFS process entailing zero wastage and hence a $MS_2$ (or $MS_{MANU}$) of 1. The last scheme on *salvaging* pertains to procurement of the extruded hollow shaft from a discarded or *end-of-life* (EOL) system, assuming it is available. Hence $MS_3$ is also 1 because of complete replacement with a salvaged shaft. Both table 1 and figure 1 follow the natural sequence in the development of a product such as a shaft. The final value of *F*, as per equation (4.1), is the sum of *S* and all the *MS* values. Consequently, the final *F* of 4, for the frugally designed shaft, is an improvement of 167% over baseline value of 1.5, due to both low *S* and various *MS* values. The rigour in applying design principles, due to a low *S*, though not apparent for this simple example is very significant for the success of the *frugal design approach*.

There are progressive savings in steel and, hence cost and energy, while sequencing through the steps of $F^S$ computation. The low *S* gives a leaner shaft with less steel, vis-à-vis a conservative design, that lowers costs and also energy consumed. For a fixed price of steel and also inexpensive manufacturing, the costs reduce in each *MS* scheme by nearly the same proportion of steel saved. So overall, there are significant savings in steel consumption and hence corresponding reductions in costs for the total *F* of 4. As for energy, assuming reasoning similar to reductions in costs results in similar savings. Therefore, this simple example attests to the large savings in materials and hence large reductions in both emissions and costs resulting from adoption of the *frugal design approach*. Moreover, equation (4.1) is a first-generation model of $F^S$ whose evolution in future would account for other *material saving* schemes and also improve its use in design.

Other than the shaft example on computation of $F^S$, quantification of just $MS_{MANU}$ for *metal cutting* is also presented, in the section on case studies, to understand its working in real time. *Metal cutting* has been selected because it is an inherently wasteful operation in terms of debris generated when compared with other processes. After this brief digression into computing $F^S$, the remainder of this paper will continue with the manufacturing aspect (related to $MS_{MANU}$) of *frugal* design.

## 4.2. Rating manufacturing process(es)

Table 2 highlights potential use of candidate processes of §3 as stand-alone FM operations. Other than zero waste, each of these processes could plausibly produce an NFS at low cost in a single step with requisite microstructure, properties, *surface integrity* and geometrical tolerances. Therefore, table 2 lists maximum $MS_{MANU}$ values, by strictly adhering to the principles of FM, for individual processes independent of any application.

However, each of the processes of table 2 and PM individually leave a finish with accompanying residual stresses and/or properties, any or all of which need improvement during actual production.

**Table 3.** Real-time *frugal manufacturing* through combinations of processes.

| no. | combinations of manufacturing processes | $MS_{MANU}$ | output |
|---|---|---|---|
| 1 | casting and metal cutting | $0 < MS_{MANU} < 1$ | net finished shape |
| 2 | additive manufacturing (AM) and metal cutting | $0 < MS_{MANU} < 1$ | net finished shape |
| 3 | powder metallurgy (PM) and heat treatment and metal cutting | $0 < MS_{MANU} < 1$ | net finished shape |
| 4 | bulk deformation and metal cutting | $0 < MS_{MANU} < 1$ | net finished shape |
| 5 | non-traditional manufacturing and metal cutting | $0 < MS_{MANU} < 1$ | net finished shape |

Table 3 therefore covers combinations of processes that could be used in real time for realizing an NFS through FM. Therefore, the primary processes of table 2 produce a NNS, under existing technology, which is subsequently transformed to an NFS through secondary operations. Consequently, $MS_{MANU}$ falls between a range of values for these process combinations while realizing an NFS, since material will have to be removed for achieving proper properties and also *surface integrity*. Other than metal removal, production of such NFSs would also entail use of surface and heat treatments with the associated cost increases for going *frugal*. Tabulations such as tables 2 and 3 should facilitate planning and implementation of any real-time FM activity.

## 4.3. Frugality from near net shape processes

After delving into individual and combinations of operations for FM, this section presents some existing instances of manufacturing that could be frugalized. Table 4 lists past efforts on NNS used in this study to make the case for FM, since these possess several requisite features. It should be noted that some of this information has been inferred where not explicitly mentioned. Overall, these NNS efforts generate parts with requisite tolerances and also proper properties but most of them lack other features of FM. As seen in table 4, none of the entries focus on *surface integrity*, which is crucial to enhancing product life and, hence, *sustainable development*. The control of *surface integrity* is important, not least for avoiding fatigue [21,22], a predominant mode of failure arising from surface flaws. This lack of focus on the quality of both surface and sub-surface regions is present even for zero wastage, which along with tolerances and properties are touted as major benefits of NNS techniques. Furthermore, a majority of these NNS entries are not single-process with several of them being expensive.

# 5. Case studies on frugal manufacturing

Four different case studies on popular manufacturing operations have been selected for demonstrating the efficacy of FM. Accordingly, table 5 lists details about these operations under relevant columns. The examples on NNS operations, i.e. superplastic forming and AM, are presented qualitatively with quantification of frugality exemplified by focusing for clarity on machining due to its intrinsic higher wastage. A *near net shape* technology can be made *frugal* since it already possesses some features

**Table 4.** Manufacturing processes from existing studies that could be 'frugalized'.

| no. | existing NNS process | tolerances | surface integrity | properties | process cost | no. processes | wastage |
|---|---|---|---|---|---|---|---|
| 1 | centrifugal casting and machining [20] | yes | no | yes | unknown | 3 | 0 |
| 2 | roll forming [32] | yes | no | yes | low | 1 | 0 |
| 3 | isothermal forging [29] | yes | no | yes | unknown | 1 | minimal |
| 4 | flow forming [23] | yes | no | yes | high | 1 | 0 |
| 5 | cold forging of spur gear [19] | yes | no | yes | unknown | 2 | minimal |
| 6 | cold forging of tulip shaft [19] | yes | no | yes | unknown | 3 | minimal |
| 7 | cold extrusion of spline shaft [41] | yes | no | yes | unknown | 4 | 0 |
| 8 | hot forging of four-stroke crank shaft [42] | yes | no | yes | low | 7 | minimal |
| 9 | superplastic forming of turbine disc [43,44] | yes | no | yes | high | 4 | 0 |

necessary for the transition. In particular, a minimal stock of feed material for producing parts with requisite shape, geometry and some properties is a step in the right direction. In this regard, entry 9 of table 4 on superplastic forming of a nickel-based superalloy, IN100, has been selected to illustrate the first case for FM. Consequently, the first entry in table 5 shows the original set of NNS processes [43,44] and also modifications for making it *frugal*. As seen in this table and figure 2, the dies used for superplastic forming could be readily retrofitted or made anew with built-in mechanism for selective heating so as to complete the heat treatment towards the end of superplastic forming. Since the die and chamber walls already possess mechanisms to heat them to superplastic temperatures, the retrofitting to control selective portions of die for post-treatment is very much possible. It should be noted that figure 2 is a general schematic of superplastic forming showing the retrofitting necessary for going *frugal* and is not a sketch of the example pertaining to turbine blades. Furthermore, the built-in selective-heating arrangement of the *frugal* process can also impart necessary residual stresses, through heat- and associated quenching-treatment [47], to sub-surface layers of the turbine. In so doing, FM does away with the post-treatments, thereby minimizing total number of processes and hence avoiding wastage of associated resources while lowering costs. In particular, when compared with separate heat treatments, cost will be reduced due to the need for only controllers that selectively heat portions of die-body in the set-up. Implicit in this argument is the realistic assumption of selecting effective and also low-cost controllers. It is noteworthy that the overall heating facilities available in the set-up could be used to combine even the *hipping* and warm-pressing operations of the conventional process, thereby leading to a single-pass arrangement.

Other than NNS technology, an important technique with significant potential to go *frugal* is AM. Accordingly, the second case study focuses on *selective laser melting* (SLM) of Ti-6Al-4V [45], a popular titanium superalloy, for depositing biomedical products. Table 5 lists the details of this AM process in the second entry, where a single pass of SLM deposits these parts. However, the resulting parts are columnar grained, possessing acicular $\alpha'$ and $\beta$ phases as opposed to the $\alpha + \beta$ phase combination characterizing a proper Ti-6Al-4V alloy, see figure 3 [45]. Thus not only do SLM-deposited Ti-6Al-4V parts comprise invalid phases, thereby leading to a different material, their grain-structure is columnar, which generally leads to severe anisotropy in properties [48]. Moreover, the invalid martensite-based acicular $\alpha'$ phase also makes these parts significantly susceptible to corrosion [45]. Although a conventional SLM approach would involve post-treatments—such as heat and/or deformation to change columnar grains to equiaxed type—to remedy the grain structure, the underlying invalid phase of acicular martensite, $\alpha'$, is difficult to rectify. Moreover, an extra finishing

**Table 5.** Case studies of processes going *frugal*. For clarity cost-reductions have been described in the main text.

| case study | manufacturing process(es) | | properties | | surface integrity | | wastage | |
|---|---|---|---|---|---|---|---|---|
| | conventional | frugal | conventional | frugal | conventional | frugal | conventional | frugal |
| *Superplastic forming* of IN100-superalloy turbine disc with blades [43,44] | hot isostatic pressing (*hipping*) to get billet warm pressing superplastic forming selective heat treatment of blades | Same as NNS same as NNS combined superplastic forming and selective heating of die portions to treat blades | coarse and fine grains in blades and disc for creep- and fatigue-strength respectively | coarse and fine grains in blades and disc for creep- and fatigue-strength respectively | does not generate proper residual stresses | can generate proper residual stresses through selective heating mechanism and quenching | minimal (see tables 2 and 3) | same as NNS |
| *selective laser melting* (SLM) of Ti-6Al-4V- superalloy for biomedical applications [45] | SLM to deposit Ti-6Al-4V part heat treatment and/or plastic deformation finish machining to get requisite *surface texture* | deposit Ti-6Al-4V alloy along with inoculant on a preheated substrate light-finish machining to get requisite *surface texture* | columnar-grain structure with atypical phases of $\alpha' + \beta$ in SLM deposition (generates different alloy with anisotropy) columnar grains are subsequently transformed to equiaxed type through heat treatment and/or plastic deformation | generates proper alloy with $\alpha + \beta$ phase combination in a single SLM pass possessing necessary equiaxed microstructure and hence proper mechanical properties | generates proper residual stresses after heat treatment and/or plastic deformation | generates proper residual stresses in a single pass | minimal (see tables 2 and 3) | minimal (see tables 2 and 3) (will be zero with futuristic advances in AM) |
| *machining* of Al7075-alloy to generate planar features for aerospace applications [46] | rough machining for maximum material removal. Depth of cut: 6 mm finish machining to obtain required finish. Depth of cut: 0.7 mm Shot peening to obtain compressive residual stresses | single cut to generate requisite finish and also compressive residual stresses (cutting speed: 1350 m min$^{-1}$. Low depth of cut: 0.7 mm). Needs less stock of feed material | proper microstructure and hence proper properties | proper microstructure and hence proper properties | mirror finish ($R_a$: between 0.5 and 0.7 μm) compressive residual stresses (between 100 and 150 MPa) | mirror finish ($R_a$: between 0.5 and 0.7 μm) compressive residual stresses (between 100 and 150 MPa) | high wastage | minimal wastage since less stock of feed material is required to begin with. $MS_{MANU} = 0.896$ |

(*Continued.*)

**Table 5.** (Continued.)

| case study | manufacturing process(es) | | properties | | surface integrity | | wastage | |
|---|---|---|---|---|---|---|---|---|
| | conventional | frugal | conventional | frugal | conventional | frugal | conventional | frugal |
| Inherently *frugal* approach to *superfinishing* from India [53]. | use of abrasive and erosion based techniques like *magnetic abrasive finishing* (MAF) and *fluidized bed assisted abrasive jet machining* (FB-AJM) respectively for generating ultra-fine finish | Simple and low-cost set-up using reciprocating action of *elastic abrasives* packed against work surface to produce ultra-fine finish | - | - | $R_a$ between 20 and 30 nm | $R_a$ between 20 and 30 nm | relatively more material removed due to heavier indenting action of grits. Possible damage to surface | no damage to surface due to flexible elastomeric action resulting in lower penetration and less material removal |
| | expensive due to necessary magnetic field source and other fixtures | abrasive grains of 23 μm size embedded on elastomeric spheres of $\phi$: 2–3 mm | | | | | | |
| | internal finishing on tubes of 440C–58HRC hardened bearing steel | internal finishing on tubes of 440C–58HRC hardened bearing steel | | | | | | |

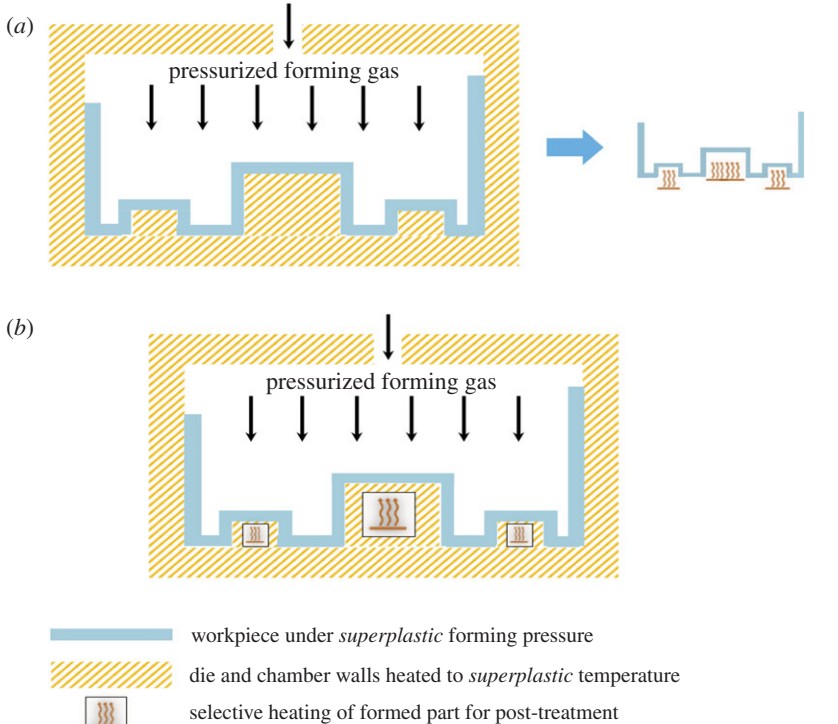

**Figure 2.** Superplastic forming as a *frugal* process. (*a*) NNS process and heat treatment, (*b*) *frugal* process incorporating heat treatment into superplastic forming set-up.

operation might also be warranted to get the requisite *surface integrity*. In contrast, as shown in figure 4, frugalizing would entail deposition of Ti-6Al-4V alloy together with an inoculant on a pre-heated substrate with the former seeding nuclei for equiaxed microstructure possessing $\alpha + \beta$ phases due to a decrease in cooling rate afforded by the latter. Moreover, a reduced thermal gradient from a preheated substrate would further suppress columnar-grain formation. The creation of a proper grain structure would also facilitate generation of necessary *residual stresses* in the part. However, a light machining cut would be required to give proper finish and, also proper residual stresses if needed, considering the current limitations of AM. Such limitations would be overcome in the foreseeable future with the current progress being made, not least on surface features of depositions from AM technologies [49,50]. Here again, the *frugal* route does away with the resources and thus costs associated with separate post-treatments. In particular, costs would be significantly reduced due to completely avoiding outlays associated with correcting both improper phases and columnar microstructure deposited by conventional SLM. Besides costs, the nearly single-pass nature also results in lower energy consumption, which is crucial since AM is considered to be energy-intensive.

The third case study is on machining, which is a workhorse of industry. Advances in plastic flow; self-excited vibratory phenomenon called chatter [34] and machine-tool systems have led to significant progress in HSM of metallic materials. As listed under the *frugal* column of table 5, aluminium AL7075 alloy can be machined in a single cut at a high speed of 1350 m min$^{-1}$ and a depth of cut of 0.7–2 mm to leave a mirror finish (Ra of 0.5 to 0.7 µm) accompanied by compressive residual stresses (100–150 MPa) [46]. As depicted in figure 5, such impressive *surface integrity* would warrant multiple passes, corresponding to a thicker stock of feed material and followed by a post-treatment of shot-peening under conventional machining involving low to moderate speeds. The use of very high cutting speeds in HSM leads to single-pass machining wherein requisite surface integrity, tolerances and properties are obtained in a single cut. So selection of a suitable depth of cut, corresponding to a thinner stock of feed material, under HSM conditions can improve productivity while leading to superior surface integrity and good properties while saving significant amounts of material and also reducing costs. In this regard, an $MS_{MANU}$ of approximately 0.9 is mentioned in the last column of table 5. Table 6 lists details of calculating this $MS_{MANU}$ through equation (4.2). The $MS_{MANU}$ of approximately 0.9 signifies savings of approximately 90% of Al 7075-T6, by adopting HSM for FM, wasted through conventional machining involving a rough and a finishing pass. Such a high value of $MS_{MANU}$ will feed into equation (4.1) and give a commensurately higher value for F of $F^S$. In other

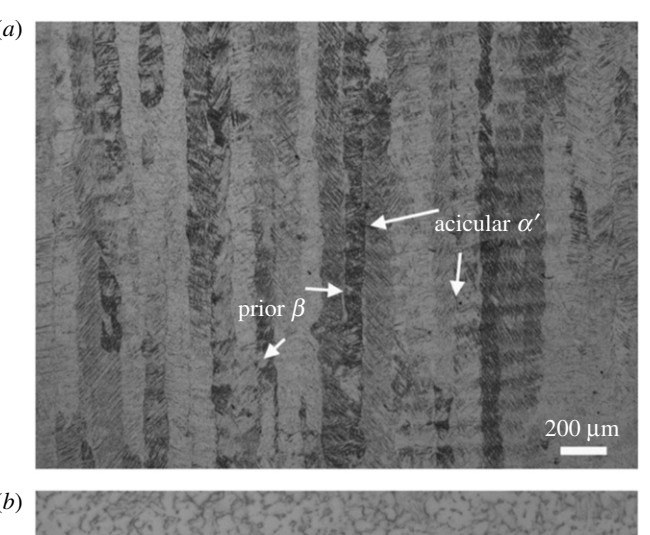

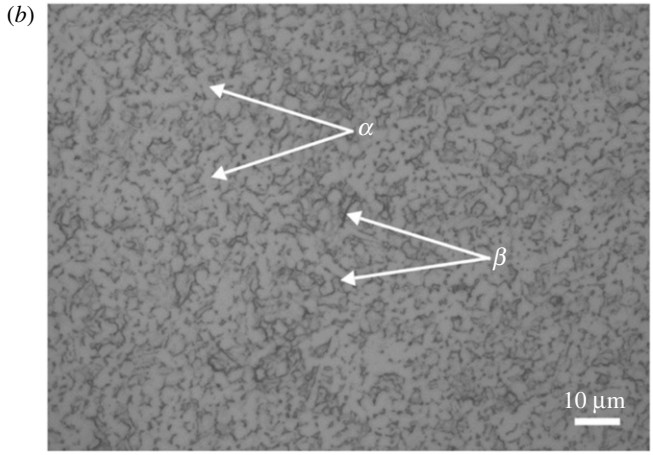

**Figure 3.** Selective laser melting of Ti-6Al-4V alloy results in undesired acicular $\alpha'$ phase along with $\beta$ in a columnar grain structure shown in (*a*). By contrast, the cast Ti-6Al-4V alloy in (*b*) possesses the desired $\alpha + \beta$ phases in an equiaxed microstructure. (Produced with permission from Zhang & Attar [45].)

words, suppressing wastage majorly through HSM enhances the frugality credentials of the concerned product. It is noteworthy that other than lesser amount of feed stock, costs in HSM can be further reduced by employing *minimum quantity lubrication* (MQL) [51] wherein a fine mist of concentrated coolant/lubricant is effectively used to significantly influence *surface integrity* and also carry away debris. Moreover, higher speeds lead to more heat generation in the deformation zone which by softening material lead to reduction of cutting forces and hence less energy consumption during HSM. Therefore, HSM is an appealing *frugal* alternative to the majority of conventional machining operations due to the resulting savings in resources, energy and costs.

The case studies covered above are inherently capital intensive. It is noteworthy to mention activities of small-scale manufacturers in India who are inherently *frugal* due to their limited revenues and lower profitability. In this regard, *The Energy and Resources Institute* (TERI) in India has studied a cluster of small-scale component manufacturers that cater to the larger *original equipment manufacturers* (OEMs) associated with the country's thriving automotive sector. TERI has encouraged this cluster in honing their potential for saving materials, reducing emissions and lowering costs. In particular, this cluster of companies in sheet-metal processing has reduced its material wastage by 15%, on average, by simply opting for better utilization of the ingoing sheet metal so as to re-use the resulting scrap as secondary raw material [52]. Therefore, the cluster has been able to achieve a cumulative $MS_{MANU}$ of 15% and a systematic application of the approach outlined in this paper can plausibly wring further savings for the companies while also improving the $F^S$ of their products. Moreover, the manufacturing research community in India is also focusing on lowering costs through alternative techniques that are better suited for frugality. An example, listed under the last entry of table 5, being the simple technique of superfinishing developed by researchers at the *Indian Institute of Space Science and Technology* (IIST) [53]. The entry in table 5 considers application of superfinishing to hardened bearing steel (440C-58HRC). This technique uses *elastic abrasives*, i.e. elastomeric balls embedded with

(*a*)

laser beam selectively
melting for deposition

meltpool

heat treatment

Ti-6Al-4V powder

finish machining

columnar grain structure and improper $\alpha' + \beta$ phases

(*b*)

finish machining

Ti-6Al-4V powder
and
inoculant

pre-heated substrate

equiaxed grain structure and proper $\alpha + \beta$ phases

**Figure 4.** Additive manufacturing as a *frugal* process. (*a*) Selective laser melting as an AM process, (*b*) selective laser melting as a *frugal* process.

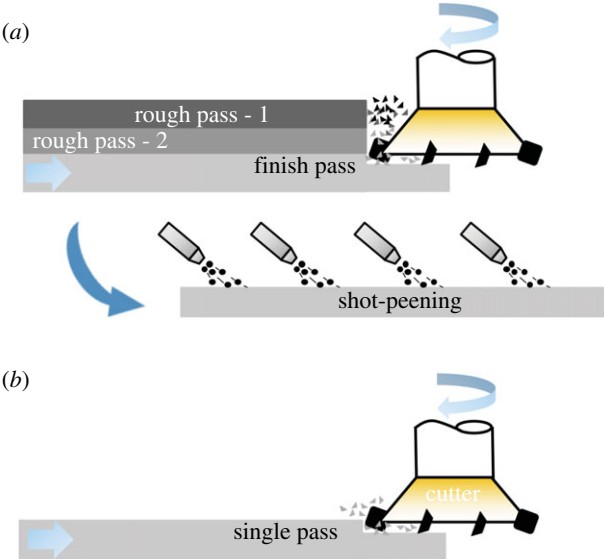

(*a*)

rough pass - 1
rough pass - 2
finish pass

shot-peening

(*b*)

cutter

single pass

**Figure 5.** High-speed machining as a *frugal* process. (*a*) Conventional machining followed by shot-peening, (*b*) high-speed machining for *frugal* cutting.

**Table 6.** $MS_{MANU}$: conventional and *frugal* machining of a rectangular block of Al 7075-T6 by uncoated carbide face-milling cutter (volume wasted = 100 mm × 70 mm × depth of cut).

| manufacturing type | depth of cut (mm) | | volume wasted (mm$^3$) | $MS_{MANU}$ |
|---|---|---|---|---|
| | rough cut | finish cut | | |
| conventional | 6 | 0.7 | 46 900 | — |
| frugal | — | 0.7 | 4900 | 0.896 |

abrasive grits, for imparting ultra-fine finish with minimal penetration of grits, thereby saving on material and also not damaging the surface. In particular, the flexible elastomeric spheres conform to the work surface by pressing against it and in the process provide large surface area with more grits that are under less pressure and hence undergo less penetration. By contrast, the conventional techniques mentioned in table 5 use expensive magnetic sources and other fixtures to arrive at such a finish with more material removal and possible surface damage due to the relatively heavier indenting action of grits. Other than simple and low-cost procedures, Indian companies are producing low-cost machine tools for all kinds of advanced manufacturing operations including those alluded to in the case studies of table 5 [54]. The use of such low-cost machine tools allows consideration of frugality even for advanced manufacturing operations. Overall, such inherently resource- and cost-saving capabilities in India and elsewhere should be tapped into to frugalize manufacturing of any kind, advanced or otherwise.

# 6. Results and discussion

A *factor of frugality*-based approach is a dedicated tool for quantifying frugality in all aspects of design including manufacturing. The savings in resources achieved through $MS_{MANU}$ feeds into improving $F^S$. Thus, the model for $MS_{MANU}$, though simplistic, forms a portion of $F^S$ and as such is crucial to the overall streamlining of a product for *frugality*. Although $W_{MAX}$, appearing in equation (4.2), might refer to hypothetical generation of maximum waste in some instances, its determination helps honing in on the actual *frugal* operation(s) that in turn lead to utmost savings in materials. In other words, selection of most wasteful real-time process(es) for $W_{MAX}$ is justified when prior numbers are unavailable. Moreover, equation (4.2) can be used for conserving materials even for manufacturing comprising wasteful operations such as *metal cutting* by aiming for a maximal value of $MS_{MANU}$. Alternatively, equation (4.2) can be used to devise recycling and reuse post-treatments necessary for putting such inevitable waste to good use.

The examples of table 5 illustrate the potential of leveraging existing processes for FM. Although most of these entries have the same wastage ($MS_{MANU}$ values), the *frugal* approach is better in terms of properties and *surface integrity* imparted to the final product with minimal number of operations. These case studies have shown potential for associated cost reductions and energy savings that are significant to fighting *climate change*. Hence, any set of arbitrary manufacturing processes could be used to do the same along the lines of table 5. Therefore, manufacturers should ideally adapt all of the existing processes including NNS techniques to being *frugal*. This initial wave of frugalization would also set the tone for discovering newer technologies that are designed for FM at inception. In fact, advances in manufacturing including robotics, automation, AI, big data, *digital twins*, *internet of things* (IoT) and the entire gamut of subtractive and additive techniques would greatly support this endeavour to go *frugal*. Such advances would greatly facilitate achieving highest $MS_{MANU}$ values while aiming for single-pass and low-cost energy-saving processes that produce quality products.

Although most of the examples in table 5 have not completely eliminated secondary operations, frugality has been instrumental in reducing their numbers. Therefore, entries of table 5 attest to FM's ability to create single-pass processes when non-*frugal* NNS operations have failed in the past [19]. Therefore, table 5 lends credence to the arguments accompanying table 2 for making stand-alone or individual processes *frugal*. This is possible, for instance, in casting operations where suitable process-conditions and moulds with smoother surfaces along with inoculants in the metal feed could be used to produce castings possessing a proper equiaxed microstructure with close tolerances and also finish. Such an equiaxed structure will also lend proper mechanical properties to the casting. AM techniques

**Table 7.** Generalized suggestions for frugalizing popular manufacturing processes

| process | machine-tool system | work material | operation | process conditions |
|---|---|---|---|---|
| metal cutting | low-cost | — | single-pass combining rough and finish cutting | higher speeds with lowest possible depth of cut |
| bulk and sheet metal | low-cost | — | single-pass involving requisite treatments; finish machining only for NNSs | finishing machining with lowest possible depth of cut |
| additive manufacturing | low-cost | addition of inoculant together with a preheated substrate for achieving proper microstructure | single-build operation followed by finish machining only for NNS-depositions | finishing machining with lowest possible depth of cut |
| casting | low-cost | addition of inoculant or other techniques for catalysing proper microstructure | single pour for moulding followed by finish machining only for NNSs | finishing machining with lowest possible depth of cut |
| non-traditional methods | low-cost | — | single-pass followed by finish machining only for NNSs | finishing machining with lowest possible depth of cut |

could be used with proper process parameters; inoculants in feed materials; pre-heated substrates and resolution-of-deposition to lay deposits that possess equiaxed microstructure with accompanying superior properties; good *surface integrity*; and hew to dimensional tolerances. The exclusion of PM techniques from table 2 is due to increasing need for post-processing such as sintering treatments to suppress porosity. Another drawback is that sintered PM parts are typically finish-machined to achieve requisite *surface integrity*. Apart from the stand-alone mode, the operations of table 2 could also be used to minimize numbers of effective processes and hence make *frugal* any arbitrary set of manufacturing processes. Also, the ideal single-pass nature of FM will generally lead to reduction of energy consumption and associated emissions.

A special focus on frugalization of AM is essential. This is due to AM's potential to produce, as a single process in a single pass, intricate shapes with equiaxed microstructure possessing superior mechanical properties that are nearly impossible to fabricate with other traditional or subtractive processes. In this regard, the ability to successfully print equiaxed microstructures for titanium- and nickel-based alloys [55,56] in a single pass should be extended to other metallic materials. Complex shapes are required for fulfilling modern engineering needs and their printing in one piece is sustainable because of the absence of assembly operations and their associated emissions. The foray of AM into *topology optimization*, wherein material is deposited only in functional regions of a product, also makes 3D printing appealing from the perspectives of both *frugality* and sustainability [57]. Also, the ability to print large objects like boats in relatively less time makes AM a good alternative to waste-prone traditional processes. Beyond depositing from scratch, techniques such as *laser engineered net shaping* (LENS^TM) can be used to repair and hence remanufacture impaired parts [48] thereby enhancing AM's salvaging and hence *frugal* abilities. Furthermore, AM is leading to novel ways of enhancing product performance such as printing of a bone-inspired architecture for substantially improving fatigue life of foam-like products. [58].

Some generalized suggestions for frugalizing the popular operations described in §3 are listed in table 7. These suggestions are one of several possible solutions for each entry. Also, a given application has to be studied in detail for their frugalization, especially in terms of properties desired

in the part/product, and these observations could temper and modify these suggestions. And the entries of table 7 are amenable to changes corresponding to progress in research on these operations. A common theme is usage of low-cost machine tools and related equipment not compromised in generating products with requisite tolerances, *surface integrity* and other properties. A single operation has also been suggested for all processes with 'treatments' referring to retrofitting machines to perform any post-treatments in tandem. As for *metal cutting*, higher speeds have been suggested along with single tool-passes for imparting better properties to fabricated surfaces.

The tenets of FM should be used to build the next generation of machine tools. However, development of *frugal machine-tool systems* (FMTSs) would have to overcome challenges rooted in the physics of manufacturing operations. An instance would be the production of NFSs through single passes of traditional forming processes possessing attendant *frugal* features encountering, *inter alia*, higher loads arising from the precision required of the operations [19]. Also, high-quality feed stock for *bulk* and *sheet* operations [19] is a must to minimize defect rates and hence facilitate successful generation of NFSs. Hence, relevant mechanics underlying existing manufacturing operations together with current advances should be used to build FMTSs from scratch or retrofit existing machine tools. In this regard, tools of Industry 4.0, such as big data, digital twins, AI etc., can be leveraged in creating FMTSs. Consequently, large outlays incurred in developing FMTSs can be recovered during usage of these machine tools due to cost savings from streamlined manufacturing operations.

Although assembly operations are ideally avoided in FM, their need in real time where necessary should be kept to a minimum. For some products, especially complicated type such as aircrafts, assembly of fabricated parts is crucial for completion of the final product. In this regard, parts designed for frugality [13,14] and made from FM processes can be assembled into a given product. Therefore, until manufacturing evolves to being low-cost single-pass operations delivering proper properties and also *surface integrity*, assembly should be designed for frugality.

FM should be universally applied to metals, non-metals and composites. This is because production technologies possessing climate-friendly credentials are significant to producers of any stripe. Consequently, the *frugal* framework can be adopted for non-metallic materials by considering relevant behaviour and also mechanics of deformation process(es) involved in their production. Accordingly, properties, *surface integrity*, shape accuracy, single pass and low cost have to be defined and selected based on the microstructure and behaviour of such materials. *3D printing* is well suited to FM of non-metallic materials and its success has been documented in recent years in depositing polymeric and also polymer-like materials for a wide range of applications including biomedical, aerospace, automotive, toy-manufacturing and others [49]. In fact, AM techniques are currently well established for polymeric and also polymer-like materials—with an intricate geometry—that do satisfy the tenets of frugality by producing parts in a single pass with good quality accompanied by minimal wastage at low cost [49,59]. Notwithstanding some issues of anisotropy, *3D printed* plastics show improved mechanical properties in certain directions vis-à-vis the traditionally injection-moulded parts [59]. Even ceramics and carbon-fibre-reinforced polymer composites have been *3D printed* recently into intricate geometries [60,61].

FM combined with the pillars of *smart factories* would buttress the foundations of sustainability. In addition to *smart factories*' tools easing adoption of frugality, the ensuing *frugal* activities would be environmentally benign and also produce quality affordable products with improved life. Therefore, futuristic *smart factories* saving on resources, energy and costs through $F^S$ would make manufacturing truly *smart*. Moreover, *smart factories*' tools and techniques would aid in designing *frugal* operations for even small manufacturers whose budgets are limited.

Apart from being rooted in sustainability, *smart factories* incorporating frugality can also have unforeseen applications, as in pandemics such as the current one involving COVID-19. In particular, FM's need for lesser numbers of workers due to its typical single-process-with-a-single-pass nature would along with amenities of *smart factories* go a long way in fighting pandemics. This is because first, the absence of secondary operations would by itself lessen opportunities for maintenance, breakdowns and other related services. And second, a smaller workforce together with cleaner facilities due to frugality and other modern technologies would avoid congestion of people and hence danger of contagion.

The starting framework being first generation covers various aspects of manufacturing and as such results in generalizations, among other limitations, which need to be addressed in subsequent efforts. The models developed are first generation that while adhering to the tenets of FM are simplistic. These models could be revised to better capture the intricacies of manufacturing. Moreover, qualitative observations need to make way for quantification as FM is adopted in specific

applications. Such quantification will also involve characterization of ensuing savings in energy and costs. Also, FM aims for savings of any kind during manufacturing and some of the indirect material-savings coming from, for instance, improvements in *surface integrity* might not be high in some cases. However, cumulative savings from a bulk of such applications would be appreciable. Similarly, smaller values of $MS_{\text{MANU}}$ may seem insignificant but their contribution over many products would give higher savings in materials and also costs.

# 7. Conclusion

Utmost reduction of resource consumption is crucial for all-round *sustainable development*, and manufacturing is an important activity for effecting this change. This paper has presented FM for generating zero waste while aiming for better part quality in terms of *surface integrity* and also mechanical and non-mechanical properties. FM also entails use of single-pass low-cost manufacturing processes for achieving these goals. These challenging demands can be met by quantifying manufacturing-frugality through a recently developed design approach based on the *factor of frugality* ($F^S$), a modern version of the *classical factor of safety* ($S$). Real-time frugalization examples have been presented on machining of an aluminium alloy, and superplastic forming and selective laser deposition of nickel- and titanium-based superalloys, respectively. Also, the inherently Indian ability of frugality has been exemplified by superfinishing of a hardened steel. Such low-cost processes and machine-tool systems can buttress the low-cost feature of FM. The potential savings in resources, energy, emissions and hence costs demonstrated by these examples, while using a minimum number of processes for producing quality parts, should be harnessed by blanket application of FM. Other than designing new-age machine-tool systems for existing and newer processes, *smart factories* should also get a fillip by harnessing frugality. By incorporating FM as another pillar, *smart factories'* efficacy in making manufacturing green would be enhanced. Moreover, a smaller workforce due to the single-process-with-single-pass nature of FM would even aid in fighting pandemics such as the current one involving COVID-19.

Data accessibility. This article has no additional data.
Competing interests. We declare we have no competing interests.
Funding. No funding has been received for this article.

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
