## [Peer Review File · Royal Society Open Science]

Review History

RSOS-210375.R0 (Original submission)

Review form: Reviewer 1

Is the manuscript scientifically sound in its present form?

Yes

Are the interpretations and conclusions justified by the results?

Yes

Is the language acceptable?

Yes

Do you have any ethical concerns with this paper?

No

Have you any concerns about statistical analyses in this paper?

No

Recommendation?

Accept with minor revision (please list in comments)

Comments to the Author(s)

The author has been active in developing a measure for characterizing the frugality of product designs. This work has been published in Open Science and other places.

In the present study, first steps are being taken to extend this frugality measure to manufacturing processes. The idea is intriguing and the necessary development is outlined. I do believe it will be a worthwhile contribution to the literature.

The author has been active in developing a measure for characterizing the frugality of product designs. This work has been published in Open Science and other places.

In the present study, the first steps are being taken to extend this frugality measure to manufacturing processes. The idea is intriguing and the necessary development is outlined. I do believe it will be a worthwhile contribution to the literature.

I would like to see the case studies better focused and developed to highlight this frugality measure. For example, the discussion of this measure based on the case studies is quite qualitative. Can the author take an example case study or two and show how this measure is established quantitatively? That would help the reader and the reviewer to better evaluate the credibility of this measure. Secondly, it will also help one understand how to walk through the measure calculation using concrete data.

It would also be helpful if these case studies pertain to some manufacturing that is intrinsically "frugal"...not case studies like high-speed machining, superplastic forming etc which are not intrinsically "frugal". For example, the machine tools required to carry out high speed machining are quite expensive. Frugality will be achieved if for example this type of machining was say eliminated by some more simpler machining technique using cheaper machine tools. Also dragging in concepts like surface integrity in the context of frugality seems a bit far-fetched. Given that the author is based in India, there should be no paucity of examples from a more indigenous "manufacturing sector" that can be used to assess this measure. Indian manufacturers are also recognized for their frugal innovations with or without availability of the proposed frugality measures. But what such an analysis with an indigenous process will do is actually enable positive changes to be realised in a local manufacturing sector. This will not only help in establishing the credibility of the proposed measure but also help it gain acceptance in the scientific manufacturing and design communities.

The aforementioned issues have to be addressed so that the validity of the frugality measure can be better assessed, and the paper will be in a form suitable for publication. I believe this can be done reasonably quickly with a better pick of the case studies.

Review form: Reviewer 2

Is the manuscript scientifically sound in its present form?

No

Are the interpretations and conclusions justified by the results?

No

Is the language acceptable?

Yes

Do you have any ethical concerns with this paper?

No

Have you any concerns about statistical analyses in this paper?

No

Recommendation?

Major revision is needed (please make suggestions in comments)

Comments to the Author(s)

This paper needs good quantification of frugality. It also needs validation. The text is verbose, repetitive, and does state obvious points. Need to go through these to eliminate a lot of unnecessary text (see Appendix A).

Decision letter (RSOS-210375.R0)

Dear Professor Rao,

The Editors assigned to your paper RSOS-210375 "FRUGAL MANUFACTURING IN SMART FACTORIES FOR WIDESPREAD SUSTAINABLE DEVELOPMENT" have now received comments from reviewers and would like you to revise the paper in accordance with the reviewer comments and any comments from the Editors. Please note this decision does not guarantee eventual acceptance.

Please submit your revised manuscript and required files (see below) no later than 21 days from today's (ie 19-Jul-2021) date. Note: the ScholarOne system will 'lock' if submission of the revision is attempted 21 or more days after the deadline. If you do not think you will be able to meet this deadline please contact the editorial office immediately.

on behalf of Professor Hazel Assender (Associate Editor) and R. Kerry Rowe (Subject Editor)
openscience@royalsociety.org

Associate Editor Comments to Author (Professor Hazel Assender):
Recommend major revisions to the manuscript, in particular changes should be made to address the need for quantification raised by both reviewers.

Reviewer comments to Author:

Reviewer: 1

Comments to the Author(s)

The author has been active in developing a measure for characterizing the frugality of product designs. This work has been published in Open Science and other places.

In the present study, first steps are being taken to extend this frugality measure to manufacturing processes. The idea is intriguing and the necessary development is outlined. I do believe it will be a worthwhile contribution to the literature.

The author has been active in developing a measure for characterizing the frugality of product designs. This work has been published in Open Science and other places.

In the present study, the first steps are being taken to extend this frugality measure to manufacturing processes. The idea is intriguing and the necessary development is outlined. I do believe it will be a worthwhile contribution to the literature.

I would like to see the case studies better focused and developed to highlight this frugality measure. For example, the discussion of this measure based on the case studies is quite qualitative. Can the author take an example case study or two and show how this measure is established quantitatively? That would help the reader and the reviewer to better evaluate the credibility of this measure. Secondly, it will also help one understand how to walk through the measure calculation using concrete data.

It would also be helpful if these case studies pertain to some manufacturing that is intrinsically "frugal"..not case studies like high-speed machining, superplastic forming etc which are not intrinsically "frugal". For example, the machine tools required to carry out high speed machining are quite expensive. Frugality will be achieved if for example this type of machining was say eliminated by some more simpler machining technique using cheaper machine tools. Also dragging in concepts like surface integrity in the context of frugality seems a bit far-fetched. Given that the author is based in India, there should be no paucity of examples from a more indigenous "manufacturing sector" that can be used to assess this measure. Indian manufacturers are also recognized for their frugal innovations with or without availability of the proposed frugality measures. But what such an analysis with an indigenous process will do is actually enable positive changes to be realised in a local manufacturing sector. This will not only help in establishing the credibility of the proposed measure but also help it gain acceptance in the scientific manufacturing and design communities.

The aforementioned issues have to be addressed so that the validity of the frugality measure can be better assessed, and the paper will be in a form suitable for publication. I believe this can be done reasonably quickly with a better pick of the case studies.

Reviewer: 2

Comments to the Author(s)

This paper needs good quantification of frugality. It also needs validation. The text is verbose, repetitive, and does state obvious points. Need to go through these to eliminate a lot of unnecessary text. (See attached file: "FrugalManufacturingReview.pdf")

===PREPARING YOUR MANUSCRIPT===

===PREPARING YOUR REVISION IN SCHOLARONE===

Author's Response to Decision Letter for (RSOS-210375.R0)

See Appendix B.

RSOS-210375.R1 (Revision)

Review form: Reviewer 1

Is the manuscript scientifically sound in its present form?

Yes

Are the interpretations and conclusions justified by the results?

Yes

Is the language acceptable?

Yes

Do you have any ethical concerns with this paper?

No

Have you any concerns about statistical analyses in this paper?

No

Recommendation?

Accept as is

Comments to the Author(s)

I m happy with the revisions.

Please fix some typographical errors (e.g., Instn Names cited in paper in lower case, Refs in garbled format etc).

Review form: Reviewer 2

Is the manuscript scientifically sound in its present form?

No

Are the interpretations and conclusions justified by the results?

No

Is the language acceptable?

Yes

Do you have any ethical concerns with this paper?

No

Have you any concerns about statistical analyses in this paper?

No

Recommendation?

Accept with minor revision (please list in comments)

Comments to the Author(s)

In summary, the author has tried to incorporate the changes suggested by the reviewer. The revised draft includes major changes in the way the manuscript is written, making it a much better draft. Thank you for doing that.

However, the lack of a detailed use-case highlighting relevant metrics w.r.t. a context, and no quantification of cost and energy savings makes it difficult to grasp the novelty of this approach apart from standard practices. 'Computation of frugality and validating them are critical to making a philosophical concept stronger,' - This forms the major deficiency in your paper. If the authors can discuss a use case to illustrate the computation of frugality, this paper will be highly referenced in the future.

Decision letter (RSOS-210375.R1)

Dear Professor Rao

On behalf of the Editors, we are pleased to inform you that your Manuscript RSOS-210375.R1 "FRUGAL MANUFACTURING IN SMART FACTORIES FOR WIDESPREAD SUSTAINABLE DEVELOPMENT" has been accepted for publication in Royal Society Open Science subject to minor revision in accordance with the referees' reports. Please find the referees' comments along with any feedback from the Editors below my signature.

Please submit your revised manuscript and required files (see below) no later than 7 days from today's (ie 15-Oct-2021) date. Note: the ScholarOne system will 'lock' if submission of the revision is attempted 7 or more days after the deadline. If you do not think you will be able to meet this deadline please contact the editorial office immediately.

on behalf of Professor Hazel Assender (Associate Editor) and R. Kerry Rowe (Subject Editor)
openscience@royalsociety.org

Associate Editor Comments to Author (Professor Hazel Assender):

Comments to the Author:

Minor revisions are required for typographical errors and referencing as required by reviewer 1.

The authors should carefully consider the further comments of reviewer 2 to further improve the manuscript. In the absence of making the changes encouraged, the manuscript should be adjusted to explicitly note the limitations/boundaries of the approach to date as described in the manuscript.

Reviewer comments to Author:

Reviewer: 1

Comments to the Author(s)

I m happy with the revisions.

Please fix some typographical errors (e.g., Instn Names cited in paper in lower case, Refs in garbled format etc).

Reviewer: 2

Comments to the Author(s)

In summary, the author has tried to incorporate the changes suggested by the reviewer. The revised draft includes major changes in the way the manuscript is written, making it a much better draft. Thank you for doing that.

However, the lack of a detailed use-case highlighting relevant metrics w.r.t. a context, and no quantification of cost and energy savings makes it difficult to grasp the novelty of this approach apart from standard practices. 'Computation of frugality and validating them are critical to making a philosophical concept stronger,' - This forms the major deficiency in your paper. If the authors can discuss a use case to illustrate the computation of frugality, this paper will be highly referenced in the future.

===PREPARING YOUR MANUSCRIPT===

While not essential, it will speed up the preparation of your manuscript proof if you format your references/bibliography in Vancouver style (please see

<https://royalsociety.org/journals/authors/author-guidelines/#formatting>). You should include DOIs for as many of the references as possible.

===PREPARING YOUR REVISION IN SCHOLARONE===

<https://royalsociety.org/journals/authors/author-guidelines/#data>. You should ensure that you cite the dataset in your reference list. If you have deposited data etc in the Dryad repository,

please only include the 'For publication' link at this stage. You should remove the 'For review' link.

Author's Response to Decision Letter for (RSOS-210375.R1)

See Appendix C.

Decision letter (RSOS-210375.R2)

Dear Professor Rao,

I am pleased to inform you that your manuscript entitled "FRUGAL MANUFACTURING IN SMART FACTORIES FOR WIDESPREAD SUSTAINABLE DEVELOPMENT" is now accepted for publication in Royal Society Open Science.

The proof of your paper will be available for review using the Royal Society online proofing system and you will receive details of how to access this in the near future from our production

office (openscience_proofs@royalsociety.org). We aim to maintain rapid times to publication after acceptance of your manuscript and we would ask you to please contact both the production office and editorial office if you are likely to be away from e-mail contact to minimise delays to publication. If you are going to be away, please nominate a co-author (if available) to manage the proofing process, and ensure they are copied into your email to the journal.

on behalf of Professor Hazel Assender (Associate Editor) and R. Kerry Rowe (Subject Editor)
openscience@royalsociety.org

Appendix A

FRUGAL MANUFACTURING IN SMART FACTORIES FOR WIDESPREAD SUSTAINABLE DEVELOPMENT

RSOS-210375

This paper describes frugal manufacturing as a fabrication scheme that utilizes a few low-cost processes for making a finished shape with desired specifications, while producing zero waste. Methods of metal manufacturing that qualify for frugal manufacturing are described. They include casting, bulk deformation and sheet processing, additive manufacturing, powder metallurgy, non-traditional techniques, and metal cutting. A factor of frugality is introduced, which is the sum of the factors of safety and material saved. The author lists manufacturing processes from existing studies that can be frugalized, which include casting, forming, among others.

1. To quote the author, “frugality is achieved in a systematic manner by starting with a low S , ~ 1.5 , design and subsequently selecting materials-and-design related features to save yet more material for this design.” The selection of S (factor of safety) depends on the design objectives, which depends on the use-case. This will be different for shaft design for power transmission components in comparison to airplane turbine blades. The notion of frugality may be secondary in many cases, where high ‘ S ’ is of primary interest. What is the incentive to start a design problem by looking at material saving (frugality)?
2. The quantification of frugality is very simplistic and requires more details for different use-cases. The use of terms like ‘proper residual stress’, ‘requisite surface texture’, ‘lesser stock’, ‘proper microstructure’, ‘high and minimal wastage’ is subjective. These terms need to be quantified to understand the engineering specifications that are being dealt with.
3. The word frugal has been used more than 150 times in the paper. Sometimes, it appears as if the focus is on advocating frugal manufacturing rather than on the problems which it can solve. How much cost and energy improvements does frugal manufacturing offer?
4. Following paragraph from the results and discussions section is very generic, and is moreover obvious, not sure what the purpose is:

“Apart from being rooted in sustainability, smart factories incorporating frugality can also have unforeseen applications as in pandemics such as the current one involving COVID-19. In particular, frugal manufacturing’s need for lesser numbers of workers due to its typical single-process-with-a single-pass nature would along with amenities of smart factories go a long way in fighting pandemics. First, the absence of secondary operations would by itself lessen opportunities for maintenance, breakdowns and other related services. Therefore, frugal operations with their lower chances of failure together with the need for a smaller workforce would help sustain manufacturing even during pandemics. And second, a smaller workforce together with cleaner facilities due to frugality, AI, robotics, digital twins, automation, modern manufacturing-operations and networking through IoT would avoid congestion of people and hence danger of contagion.”

5. Following paragraph from the ‘Why frugality in smart factories’ section is also very generic. There are a lot of keywords without any details. The author describes frugality as efforts towards high quality, zero waste, and low-cost (minimal number of operations). However,

these factors have been considered in manufacturing for a long time, for example in lean manufacturing.

6. "In fact, building of novel machine-tools, designed from the perspective of frugal manufacturing, would be greatly aided by the pillars of Industry 4.0, ie, big data and simulation techniques involving digital twins and artificial intelligence (AI). All in all, incorporating frugal manufacturing into Industry 4.0 will make factories truly "smart" by "frugalizing" their process(es) against climate-change." – not sure what the purpose is?
7. In Section 3, Manufacturing of metals was considered, and few specific processes were picked. Was there any reason behind their choices? Was there some metrics to pick or how did these qualify for frugal manufacturing?
8. Maybe some pointers on "how the process in Section 3 could be made frugal" can be helpful to the readers to get an idea about this concept. Can list the transformation in a table for quick summary.
9. MS_{MANU} in Section 4.1 quantifies material wasted in manufacturing; So, in frugal manufacturing shouldn't this be minimized? These equations can be made clearer.
10. Are there any limitations with respect to applying frugal manufacturing? A section on this can be added.
11. Can the factor of frugality be computed for Case Studies? Adding that might serve as a validation and will give readers an idea of how to use the different parameters.
12. Tables seem to be scattered around in the paper. It might be better to place tables close to the paragraphs where they are referred.
13. Name of table needs to be placed above the table.

In summary, the paper introduces the concept of frugal manufacturing which emphasizes on waste minimization. Some opportunities of leveraging frugality in metal manufacturing are presented, along with three use-cases. The metrics defining frugality and its characteristics need to be more detailed. The motivation of using a frugality-based approach is not clear from the text.

Quantification of subjective terms is needed to better understand the improvement opportunities. The write-up can be made concise, while limiting the repetitive usage of some key words.

Computation of frugality and validating them are critical to making a philosophical concept stronger.

Appendix B

Response to Referees

Title: FRUGAL MANUFACTURING IN SMART FACTORIES FOR WIDESPREAD SUSTAINABLE DEVELOPMENT

I would like to thank the editor and reviewers for their positive and valuable comments on my research effort. I also thank them for taking the time to point out meticulously issues pertaining to the content of this work. I have accordingly tried to answer the various queries and also revise the initial manuscript. Please find answers to the questions raised by reviewers in the lines that follow. The edited portions appear underlined here and red-colored in the revised manuscript.

Associate Editor

1. Recommend major revisions to the manuscript, in particular changes should be made to address the need for quantification raised by both reviewers.

ANS: I thank the editor for this comment. I have attempted to answer each of the questions raised by the reviewers and accordingly modify the initial manuscript. In doing so, I have quantified in the revised manuscript MS_{MANU} , which goes into computing *factor of frugality* (F^S), for the existing case study on machining, which is an inherently wasteful operation. Details on quantification of F^S have already appeared in this journal and elsewhere [1-2] and the interested reader can refer to these references that have been mentioned in my manuscript. I have used the qualitative validations for the remaining existing studies on forming and additive manufacturing, which are suitable for *near net shape* processes. Also, this is a first generation effort that sheds light more on the framework or philosophy behind *frugal manufacturing* with futuristic efforts possibly shedding more light on quantification through undertaking of specific applications.

Reviewer 1

1. The author has been active in developing a measure for characterizing the frugality of product designs. This work has been published in Open Science and other places.

In the present study, first steps are being taken to extend this frugality measure to manufacturing processes. the idea is intriguing and the necessary development is outlined. I do believe it will be a worthwhile contribution to the literature.

The author has been active in developing a measure for characterizing the frugality of product designs. This work has been published in Open Science and other places. In the present study, the first steps are being taken to extend this frugality measure to manufacturing processes. The idea is intriguing and the necessary development is outlined. I do believe it will be a worthwhile contribution to the literature.

ANS: I thank the reviewer for this comment.

2. I would like to see the case studies better focused and developed to highlight this frugality measure. For example, the discussion of this measure based on the case studies is quite qualitative. Can the author take an example case study or two and show how this measure is established quantitatively? That would help the reader and the reviewer to better evaluate the credibility of this measure. Secondly, it will also help one understand how to walk through the measure calculation using concrete data.

ANS: I thank the reviewer for this suggestion and I do agree about the qualitative nature of some of these case studies. I have attempted to, at the very minimum, demonstrate that *frugal manufacturing* (FM) covers nearly all types of processes including forming, additive (variant of casting) and cutting. Forming and additive manufacturing being *near net shape processes* involve relatively lesser wastage from possible finish-machining and so emphasis has been placed on qualitatively comparing other FM-tenets of lesser numbers of processes and quality.

As per the reviewer's suggestion, in the revised version, I have selected the machining operation from the reported case studies because it typically has the highest wastage and hence can clarify the frugality approach better. Accordingly, Table 5 presented below and, also included in the revised manuscript, brings out the material-savings on opting for single-pass *high-speed-machining* (HSM) in place of the conventional type that typically comprises separate rough and finishing cuts involving higher volumes of metal removal. The MS_{MANU} together with some other details is also listed in Table 4 for the case study on HSM. The following lines have also been added under relevant sections:

Section 4.1 (last paragraph): “An example showing the computation of MS_{MANU} for metal cutting, along with its implications for F^S , is presented in the section on case studies. Metal cutting has been selected because it is an inherently wasteful operation in terms of debris generated when compared to NNS processes.”

Section 5 (first paragraph): “The examples on NNS operations, i.e., superplastic forming and additive manufacturing, are presented qualitatively with quantification of frugality exemplified by focusing for clarity on machining due to its intrinsic higher wastage.”

Section 5 (third paragraph): “In this regard, an MS_{MANU} of ~0.9 is mentioned in the last column of Table 4. Table 5 lists details of calculating this MS_{MANU} through equation (2). The MS_{MANU} of ~0.9 signifies savings of ~90% of Al 7075-T6, by adopting HSM for FM, that is wasted through conventional machining involving a rough and a finishing pass. Such a high value of MS_{MANU} will feed into equation (1) and give a commensurately higher value for F of F^S . In other words, suppressing wastage majorly through HSM enhances the frugality credentials of the concerned product.”

Table 5: MS_{MANU} : Conventional and *frugal* machining of a rectangular block of Al 7075-T6 by uncoated carbide face-milling cutter
(Volume wasted = 100mm x 70mm x Depth of Cut)

Manufacturing Type	Depth of Cut (mm)		Volume Wasted (mm ³)	MS _{MANU}
	Rough Cut	Finish Cut		
Conventional	6	0.7	46900	-
Frugal	-	0.7	4900	0.896

3. It would also be helpful if these case studies pertain to some manufacturing that is intrinsically "frugal"..not case studies like high-speed machining, superplastic forming etc which are not intrinsically "frugal". For example, the machine tools required to carry out *high speed machining* are quite expensive. Frugality will be achieved if for example this type of machining was say eliminated by some more simpler machining technique using cheaper machine tools. Also dragging in concepts like surface integrity in the context of frugality seems a bit far-fetched. Given that the author is based in India, there should be no paucity of examples from a more indigenous "manufacturing sector" that can be used to assess this measure. Indian manufacturers are also recognized for their frugal innovations with or without availability of the proposed frugality measures. But what such an analysis with an indigenous process will do is actually enable positive changes to be realised in a local manufacturing sector. This will not only help in establishing the credibility of the proposed measure but also help it gain acceptance in the scientific manufacturing and design communities.

ANS: I thank the reviewer for sharing these insights and related suggestions that have helped in extending the range of applicability and clarify concepts in the revised manuscript. This being a first attempt on establishing frugality in manufacturing of metals, I have tried to be quite broad because this concept has the potential to influence both advanced and alternative manufacturing techniques. There will be

different degrees of frugality between the ones presented in my paper and the inherently-frugal ones that the reviewer is alluding to. Since metals-based manufacturing is to some extent inherently expensive, even smaller savings in individual cases can add up when considering higher volumes of such operations. I have added a new paragraph on the inherently *frugal* way of manufacturing in India under case studies. This together with existing case studies in the revised manuscript forms a suitable set of examples for “frugalization” of manufacturing. Moreover, the Indian angle on *frugal manufacturing* of metals is a new topic needing time that I will be actively pursuing in future research for the adoption and use of *frugal manufacturing* in India.

A new paragraph has been added under case studies in the revised manuscript that describes a study carried out by *the energy and resources institute* (TERI) in India for honing existing *frugal* potential in small scale manufacturers of automotive parts. I have also added brief implications of using the frugality measure to achieve this and more systematically. Other than the small scale segment, a research effort from the *Indian Institute of Space Science and Technology* (IIST) has also been presented in this paragraph to show the inherently *frugal* potential of researchers from India’s manufacturing community in carrying out sophisticated work. This work listed as a new entry under Table 4 (listed below) uses *elastic abrasives*, i.e., elastomeric beads with abrasive grits, that are pressed and reciprocated against internal tubular surfaces for carrying out ultra fine finishing frugally. In doing so, *Sooraj & Radhakrishnan* [3] have ingeniously avoided expensive magnetic sources and other fixtures of traditional techniques listed in Table 4 while achieving same or better finish.

The Indian manufacturing sector has always shown a propensity for cutting costs that continues to this day. Even with the economic growth seen over the past decades, this sector has continued to pursue all kinds of manufacturing activities, including sophisticated ones like *high speed machining*, at lower costs. So lastly, I have added lines highlighting the lower costs of Indian machine tools that are used for even sophisticated tasks like high-speed-machining. Overall, the following lines have been added as a last paragraph under the section on case studies together with a new entry under Table 4:

Section 5 (last paragraph): “The case studies covered above are inherently capital intensive. It is noteworthy to mention activities of small-scale manufacturers in India

who are inherently *frugal* due to their limited revenues and lower profitability. In this regard, *the energy and resources institute* (TERI) in India has studied a cluster of small scale component manufacturers that cater to the larger *original equipment manufacturers* (OEMs) associated with the country's thriving automotive sector. TERI has encouraged this cluster in honing their potential for saving materials, reducing emissions and lowering costs. In particular, this cluster of companies in sheet-metal-processing has reduced its material wastage by 15%, on average, by simply opting for better utilization of the ingoing sheet metal so as to reuse the resulting scrap as secondary raw material [4]. Therefore, the cluster has been able to achieve a cumulative MS_{MANU} of 15% and a systematic application of the approach outlined in this paper can plausibly wring further savings for the companies while also improving the F^S of their products. Moreover, the manufacturing-research community in India is also focusing on lowering costs through alternate techniques that are better suited for frugality. An example, listed under the last entry of Table 4, being the simple technique of superfinishing developed by researchers at the *Indian institute of space science and technology* (IIST) [3]. The entry in Table 4 considers application of superfinishing to hardened bearing steel (440C-58HRC). This technique uses *elastic abrasives*, i.e., elastomeric balls embedded with abrasive grits, for imparting ultra fine finish with minimal penetration of grits thereby saving on material and also not damaging the surface. In particular, the flexible elastomeric spheres conform to the work surface by pressing against it and in the process provide large surface area with more grits that are under lesser pressure and hence undergo lesser penetration. In contrast, the conventional techniques mentioned in Table 4 use expensive magnetic sources and other fixtures to arrive at such a finish with more material removal and possible surface-damage due to the relatively heavier indenting action of grits. Other than simple and low-cost procedures, Indian companies are producing low-cost machine tools for all kinds of advanced manufacturing operations including those alluded to in the case studies of Table 4 [5]. The use of such low-cost machine tools allow considerations of frugality even for advanced manufacturing operations. Overall, such inherently resource-and-cost saving capabilities in India and elsewhere should be tapped into to frugalize manufacturing of any kind, advanced or otherwise.”

As for global activity, I do agree with the reviewer that manufacturing as currently practiced involves machine tools and supporting technologies that make it expensive. However, my effort attempts to set a framework for “frugalizing” any manufacturing activity irrespective of its underlying technology. Hence, this work attempts to identify and control any aspect of manufacturing including product-quality in terms of *surface integrity* for achieving savings in materials and hence costs. *Surface integrity* was invoked here because its improvement, at least in terms of proper residual stresses, can enhance the life of an individual product and hence indirectly contribute to material savings and cost. Small savings in *surface integrity* would plausibly add on over many products to contribute to the maximum possible for any given manufacturing activity. Similarly, many such smaller savings over other aspects of manufacturing would also plausibly contribute to decent savings for an individual product that would later amplify through mass production. I have added a couple of lines on *surface integrity* in the last paragraph under “Results & Discussion” on limitations of my approach. These selected lines are:

Section 6 (last paragraph): “Also, FM aims for savings of any kind during manufacturing and some of the indirect material-savings coming from, for instance, improvements in *surface integrity* might not be high in some cases. However, cumulative savings from a bulk of such applications would be appreciable. Similarly, smaller values of MS_{MANU} may seem insignificant but their contribution over many products would give higher savings in materials and also costs.”

Table 4 (new entry): Case studies of processes going frugal.

Case Study	Manufacturing Process(es)		Properties		Surface Integrity		Wastage	
	Conventional	Frugal	Conventional	Frugal	Conventional	Frugal	Conventional	Frugal
Inherently frugal approach to super-finishing from India [3]	Use of abrasive and erosion based techniques like Magnetic Abrasive Finishing (MAF) and Fluidized Bed assisted Abrasive Jet Machining (FB-AJM) respectively for generating ultra fine finish. Expensive due to necessary magnetic field source and other fixtures. Internal finishing on tubes of 440C-58HRC hardened bearing steel.	Simple and low-cost setup using reciprocating action of elastic abrasives packed against work surface to produce ultra fine finish. Abrasive grains of 23 μm size embedded on elastomeric spheres of ϕ: 2-3 mm. Internal finishing on tubes of 440C-58HRC hardened bearing steel.	-	-	R_a between 20 and 30 nm	R_a between 20 and 30 nm	Relatively more material removed due to heavier indenting action of grits. Possible damage to surface.	No damage to surface due to flexible elastomeric action resulting in lower penetration and lesser material removal.

The aforementioned issues have to be addressed so that the validity of the frugality measure can be better assessed, and the paper will be in a form suitable for publication. I believe this can be done reasonably quickly with a better pick of the case studies.

ANS: The reviewer's suggestions have been very useful in arriving at the revised manuscript. I have attempted to modify the manuscript accordingly and the revised manuscript has accommodated many of these changes.

Reviewer 2

1. This paper needs good quantification of frugality. It also needs validation. The text is verbose, repetitive, and does state obvious points. Need to go through these to eliminate a lot of unnecessary text. (See attached file: "FrugalManufacturingReview.pdf")

ANS: I thank the reviewer for these general comments. It reads verbose because this is also a starting framework that attempts to frame all the concepts. I believe all of this content is required because *frugal manufacturing* touches upon so many of the existing concepts in manufacturing. However, as mentioned in subsequent portions of this review I have culled some unnecessary portions to make the reading better. Many other redactions, not mentioned in this document, to reduce the verbosity are also marked in the revised manuscript in red color. The reviewer's suggestions have been very helpful in this regard.

For quantification of MS_{MANU} the case study on *high speed machining* has been selected in the revised version to clearly bring out the working of the frugality approach. Accordingly, the following together with Table 5 have been added in the revised manuscript.

Section 4.1 (last paragraph): “An example showing the computation of MS_{MANU} for metal cutting, along with its implications for F^S , is presented in the section on case

studies. Metal cutting has been selected because it is an inherently wasteful operation in terms of debris generated when compared to NNS processes.”

Section 5 (first paragraph): “The examples on NNS operations, i.e., superplastic forming and additive manufacturing, are presented qualitatively with quantification of frugality exemplified by focusing for clarity on machining due to its intrinsic higher wastage.”

Section 5 (third paragraph): “In this regard, an MS_{MANU} of ~ 0.9 is mentioned in the last column of Table 4. Table 5 lists details of calculating this MS_{MANU} through equation (2). The MS_{MANU} of ~ 0.9 signifies savings of $\sim 90\%$ of Al 7075-T6, by adopting HSM for FM, wasted through conventional machining involving a rough and a finishing pass. Such a high value of MS_{MANU} will feed into equation (1) and give a commensurately higher value for F of F^S . In other words, suppressing wastage majorly through HSM enhances the frugality credentials of the concerned product.”

Table 5: MS_{MANU} : Conventional and *frugal* machining of a rectangular block of Al 7075-T6 by uncoated carbide face-milling cutter

(Volume wasted = 100mm x 70mm x Depth of Cut)

Manufacturing Type	Depth of Cut (mm)		Volume Wasted (mm ³)	MS_{MANU}
	Rough Cut	Finish Cut		
Conventional	6	0.7	46900	-
Frugal	-	0.7	4900	0.896

As for F^S , the quantification details have been covered under previous publications [1-2] in this journal and elsewhere.

2. This paper describes frugal manufacturing as a fabrication scheme that utilizes a few low-cost processes for making a finished shape with desired specifications, while producing zero waste. Methods of metal manufacturing that qualify for frugal manufacturing are described. They include casting, bulk deformation and sheet processing, additive manufacturing, powder metallurgy, non- traditional techniques,

and metal cutting. A factor of frugality is introduced, which is the sum of the factors of safety and material saved. The author lists manufacturing processes from existing studies that can be frugalized, which include casting, forming, among others.

To quote the author, “frugality is achieved in a systematic manner by starting with a low S , ~ 1.5 , design and subsequently selecting materials-and-design related features to save yet more material for this design.” The selection of S (factor of safety) depends on the design objectives, which depends on the use-case. This will be different for shaft design for power transmission components in comparison to airplane turbine blades. The notion of frugality may be secondary in many cases, where high ‘ S ’ is of primary interest. What is the incentive to start a design problem by looking at material saving (frugality)?

ANS: I thank the reviewer for this query. Yes, the S value will depend on a given design case. However, even for a case with a high value of S the *frugal* approach can be pursued to save materials and costs. Such savings will amplify when frugality is achieved over many high- S products. Against the backdrop of sustainability, any savings in materials is desired since that is better from an environmental point of view. And costs will also typically be saved which is important for commercial entities. In fact, pursuing *frugal design* with cutting-edge research will eventually aid in getting to low- S designs even for current cases where a high S is used.

3. The quantification of frugality is very simplistic and requires more details for different use- cases. The use of terms like ‘proper residual stress’, ‘requisite surface texture’, ‘lesser stock’, ‘proper microstructure’, ‘high and minimal wastage’ is subjective. These terms need to be quantified to understand the engineering specifications that are being dealt with.

ANS: I do agree that quantification appears simplistic and in particular MS_{MANU} characterizes savings in weight of wastage from conventional manufacturing operations. But it captures the gist of saving materials through manufacturing of any kind where a lot gets wasted in arriving at finished parts/products. However, such simplistic terms (from manufacturing and other aspects) subsequently contribute cumulatively to F^S and hence simplicity also helps in better control of F^S through

various MS schemes. Moreover, these are first generation models with possible modifications coming in the foreseeable future.

The quantification of terms will depend on applying this approach to specific applications. However, the case study on *high speed machining* mentions specific values for roughness and residual stresses that are required of the machined surface. Overall, this paper sets the framework for “frugalizing” manufacturing and hence aims for minimal wastage against better quality in terms of the features mentioned in this question. Any application has to be judged against these terms and will acquire specific values relevant to that application. Subsequent research efforts from my group and others would attempt to undertake applications.

I have added lines to reflect these issues in the revised manuscript under limitations of this approach in the last paragraph of section 6. Details can be found in my answer to Q 11.

4. The word frugal has been used more than 150 times in the paper. Sometimes, it appears as if the focus is on advocating frugal manufacturing rather than on the problems which it can solve. How much cost and energy improvements does frugal manufacturing offer?

ANS: Sorry about this usage, which I do agree is avoidable. The appearance of the *frugal* term has been corrected by avoiding it in many places. Plus, *frugal manufacturing's* abbreviation of FM has been used in several places to avoid increased used of *frugal* term.

Since this is a first generation effort for setting up the FM-framework, it has qualitative assessments on energy and cost savings among others. These will need to be addressed in subsequent efforts when the theory underlying FM will be adopted in specific applications. Accordingly I have added the following under “Results and Discussion” in the last paragraph on limitations:

Section 6 (relevant portions of last paragraph): “The starting framework being first generation covers various aspects of manufacturing and as such results in generalizations, among other limitations, which need to be addressed in subsequent efforts. The models developed are first generation that while adhering to the tenets of FM are simplistic. These models could be revised to better capture the intricacies of manu-

facturing. Moreover, qualitative observations need to make way for quantification as FM is adopted in specific applications. Such quantification will also involve characterization of ensuing savings in energy and costs.”

5. Following paragraph from the results and discussions section is very generic, and is moreover obvious, not sure what the purpose is: “Apart from being rooted in sustainability, smart factories incorporating frugality can also have unforeseen applications as in pandemics such as the current one involving COVID-19. In particular, frugal manufacturing’s need for lesser numbers of workers due to its typical single- process-with-a single-pass nature would along with amenities of smart factories go a long way in fighting pandemics. First, the absence of secondary operations would by itself lessen opportunities for maintenance, breakdowns and other related services. Therefore, frugal operations with their lower chances of failure together with the need for a smaller workforce would help sustain manufacturing even during pandemics. And second, a smaller workforce together with cleaner facilities due to frugality, AI, robotics, digital twins, automation, modern manufacturing-operations and networking through IoT would avoid congestion of people and hence danger of contagion.”

ANS: Again, since this work has taken the first steps in presenting *frugal manufacturing* the presentation while appearing generic is trying to cover all areas of impact. I have culled and done a replacement to keep this paragraph short:

Section 6: (lines culled in paragraph 10): “Therefore, frugal operations with their lower chances of failure together with the need for a smaller workforce would help sustain manufacturing even during pandemics.”

Section 6: (lines culled and replaced in paragraph 10): “...AI, robotics, digital twins, automation, modern manufacturing-operations and networking through IoT...”
has been replaced with “...other modern technologies...”

6. Following paragraph from the ‘Why frugality in smart factories’ section is also very generic. There are a lot of keywords without any details. The author describes frugality as efforts towards high quality, zero waste, and low-cost (minimal number of

operations). However, these factors have been considered in manufacturing for a long time, for example in lean manufacturing.

ANS: I thank the reviewer for bringing in the lean concept. *Frugal manufacturing* subsumes “lean” (or minimum-number-of-operations) and goes beyond that by placing emphasis on product-quality and costs. Up until now, as the reviewer suggests, high quality, zero waste and low-cost have been considered in manufacturing, but FM brings all three together. To my knowledge, this has not been done before.

Moreover, aiming collectively for these three features of FM is important to the *smart factory* concept due to FM’s cost-effective ability to streamline operations while also aiming for quality parts. In other words, FM would facilitate enhancing efficiency in manufacturing which was the motive for starting the *smart factory* concept. I have tried to keep this paragraph as limited as possible in the initial draft.

7. “In fact, building of novel machine-tools, designed from the perspective of frugal manufacturing, would be greatly aided by the pillars of Industry 4.0, ie, big data and simulation techniques involving digital twins and artificial intelligence (AI). All in all, incorporating frugal manufacturing into Industry 4.0 will make factories truly ”smart” by “frugalizing” their process(es) against climate-change.” – not sure what the purpose is?

ANS: The answer to this query is a continuation of the answer to the previous question. Also, since FM relates to actual operations and hence their relevant physics, it will play a major role in streamlining operations for resource and emission control. Industry 4.0 does not directly influence operations and hence adding FM to its mix of tools and concepts will strengthen this new standard, which is being widely adopted by manufacturers.

As for building of novel machine tools for frugality, this would require the usage of existing tools used in Industry 4.0. This is because achieving the three goals of FM together would require not least, assimilation of relevant data and also observations based on mechanics underlying the processes. This would be effectively achieved

through various interactions between these new machine tools and relevant technologies of Industry 4.0.

Hence, while FM's inclusion will strengthen Industry 4.0, the existing tool-kit (i.e., without FM) of Industry 4.0 is needed to realize FM in real-time *smart factories*. Accordingly, I have added the following sentence:

Section 2.1: “Inclusion of FM in Industry 4.0 is essential since FM controls actual operations based on their relevant physics, which Industry 4.0 does not. But realization of FM in real-time *smart factories* will require, inter alia, interactions between machine-tool systems and existing tool-kit of Industry 4.0 including digital twins and *artificial intelligence* (AI), to name a few.”

Section 2.1 (this portion has been culled): “.....ie, big data and simulation techniques involving digital twins and artificial intelligence (AI).”

Paragraph 6 of “Results and Discussion” section is also devoted to the new machine tool systems and gives more explanation. I have also added the following sentence in that paragraph to add clarity:

Section 6 (paragraph 6): “In this regard, tools of Industry 4.0, such as *big data*, digital twins, AI etc., can be leveraged in creating FMTs.”

8. In Section 3, Manufacturing of metals was considered, and few specific processes were picked. Was there any reason behind their choices? Was there some metrics to pick or how did these qualify for frugal manufacturing?

ANS: This paper is a first one on *frugal manufacturing* and I have limited it to metallic materials due to their widespread use and also because their science-and-engineering is fairly well developed. The operations selected are fundamental and also popular in the metals-based-industry where these processes or their variants are used. Future efforts in FM will be directed toward non-metals, hybrids and others.

9. Maybe some pointers on "how the process in Section 3 could be made frugal" can be helpful to the readers to get an idea about this concept. Can list the transformation in a table for quick summary.

ANS: I thank the reviewer for this pointer and have attempted to include a new table (#6) under the section on “Results and Discussion”. It should be noted that these suggestions are general and frugalization has to be undertaken relevant to a given application. I have accordingly added the following lines under a new paragraph in this section:

Section 6 (Paragraph 5): “Some generalized suggestions for frugalizing the popular operations described in section 3 are listed in Table 6. These suggestions are one of several possible solutions for each entry. Also, a given application has to be studied in detail for their frugalization, especially in terms of properties desired in the part/product and, these observations could temper and modify these suggestions. And, the entries of Table 6 are amenable to changes corresponding to progress in research on these operations. A common theme is usage of low-cost machine tools and related equipment not compromised in generating products with requisite tolerances, *surface integrity* and other properties. A single operation has also been suggested for all processes with “treatments” referring to retrofitting machines to perform any post treatments in tandem. As for *metal cutting*, higher speeds have been suggested along with single tool-passes for imparting better properties to fabricated surfaces.”

The table added is listed below:

Table 6: Generalized suggestions for frugalizing popular manufacturing processes

Process	Machine-Tool System	Work Material	Operation	Process Conditions
Metal Cutting	Low-cost	-	Single-pass combining rough and finish cutting	Higher speeds with lowest possible depth of cut
Bulk & Sheet Metal	Low-cost	-	Single-pass involving requisite treatments. Finish	Finishing machining with lowest possible depth of

			machining only for NNSs	cut
Additive Manufacturing	Low-cost	Addition of inoculant together with a preheated substrate for achieving proper microstructure	Single build operation followed by finish machining only for NNS-depositions	Finishing machining with lowest possible depth of cut
Casting	Low-cost	Addition of inoculant or other techniques for catalyzing proper microstructure	Single pour for molding followed by finish machining only for NNSs	Finishing machining with lowest possible depth of cut
Non Traditional Methods	Low-cost	-	Single-pass followed by finish machining only for NNSs	Finishing machining with lowest possible depth of cut

10. MS_{MANU} in Section 4.1 quantifies material wasted in manufacturing; So, in frugal manufacturing shouldn't this be minimized? These equations can be made clearer.

ANS: MS_{MANU} determines the savings in waste coming from a *frugal* operation when compared to the conventional type. So these savings and hence MS_{MANU} has to be maximized. The revised manuscript gives the computation for the case study on machining where MS_{MANU} was calculated to be 0.895 (~0.9) meaning 89.5% (~90%) of waste coming from a conventional manufacturing operation was saved by opting for the *frugal* one of *high speed machining*. Therefore, MS_{MANU} has to be maximized for saving maximum amount of debris coming from a conventional operation.

11. Are there any limitations with respect to applying frugal manufacturing? A section on this can be added.

ANS: I have added a paragraph at the end of “Results and Discussion” on some limitations:

Section 6 (last paragraph): “The starting framework being first generation covers various aspects of manufacturing and as such results in generalizations, among other limitations, which need to be addressed in subsequent efforts. The models developed are first generation that while adhering to the tenets of FM are simplistic. These models could be revised to better capture the intricacies of manufacturing. Moreover, qualitative observations need to make way for quantification as FM is adopted in specific applications. Such quantification will also involve characterization of ensuing savings in energy and costs. Also, FM aims for savings of any kind during manufacturing and some of the indirect material-savings coming from, for instance, improvements in *surface integrity* might not be high in some cases. However, cumulative savings from a bulk of such applications would be appreciable. Similarly, smaller values of MS_{MANU} may seem insignificant but their contribution over many products would give higher savings in materials and also costs.”

12. Can the factor of frugality be computed for Case Studies? Adding that might serve as a validation and will give readers an idea of how to use the different parameters.

ANS: The *factor of frugality* (F^S) is computed overall for a given product design and this paper deals with only one of the *material saved* (MS) parameters going into its computation that is associated with FM, i.e., MS_{MANU} . Therefore, MS_{MANU} plus other MS parameters for other aspects of product development together with a low factor of safety go into the computation of F^S . A brief explanation on evaluating F^S has been described in section 4.1 while more details, with computations, can be found in my earlier papers in this journal [1] and elsewhere [2]. These references were mentioned in the initial manuscript.

However, the computation of MS_{MANU} has been reported for the case study on *metal cutting* in the revised manuscript. This has been done to add clarity to the actual usage of MS_{MANU} . The details about this addition can be found in the answer to first question. I have also added the following sentence for clarity:

Section 4.1: “The MS parameter associated with manufacturing, i.e., FM, is denoted by MS_{MANU} .”

13. Tables seem to be scattered around in the paper. It might be better to place tables close to the paragraphs where they are referred.

ANS: The tables have been now arranged closer to the text in the revised manuscript.

14. Name of table needs to be placed above the table.

ANS: The captions have been placed above the tables.

15. In summary, the paper introduces the concept of frugal manufacturing which emphasizes on waste minimization. Some opportunities of leveraging frugality in metal manufacturing are presented, along with three use-cases. The metrics defining frugality and its characteristics need to be more detailed. The motivation of using a frugality-based approach is not clear from the text. Quantification of subjective terms is needed to better understand the improvement opportunities. The write-up can be made concise, while limiting the repetitive usage of some key words.

Computation of frugality and validating them are critical to making a philosophical concept stronger.

ANS: The reviewer’s suggestions have been very useful in arriving at the revised manuscript. The details defining the metrics of frugality have already appeared in earlier publications in this journal and elsewhere [1-2]. This paper deals with the details of one engineering aspect, i.e., manufacturing, going into the computation of the *factor of frugality*. Hence, one of the MS parameters, i.e., MS_{MANU} , needed for computing the *factor of frugality* is partially the topic of interest. Moreover, the motivation for using a frugality based approach has been described in this work and my earlier efforts [1-2] have more details on it. Also, since this paper presents the

framework or philosophy of FM, quantification of the subjective portions will be topics of subsequent efforts.

References

- [1] Rao, B. C. (2019), 'The science underlying frugal innovations should not be frugal', *Royal Society Open Science*, Vol. 6(5), 180421.
- [2] Rao, B. C. (2017b), 'Revisiting classical design in engineering from a perspective of frugality', *Heliyon*, Vol.3(5), e00299.
- [3] Sooraj, V. and Radhakrishnan, V. (2014), 'Fine finishing of internal surfaces using elastic abrasives', *International Journal of Machine Tools Manufacture*, Vol. 78, pp. 30–40.
- [4] TERI (2016), in 'Towards a resource efficient auto component manufacturing in India: A case study', Workshop on Towards a Resource Efficient Auto Component Manufacturing in India.
- [5] UNIDO (2005), National programme for development of the machine tool industry in india: A success story, Technical report, United Nations Industrial Development Organization (UNIDO), Vienna.

Appendix C

Response to Referees

Title: FRUGAL MANUFACTURING IN SMART FACTORIES FOR WIDESPREAD SUSTAINABLE DEVELOPMENT

I would like to thank the editor and reviewers for their valuable comments on the revised manuscript. I also thank them for taking the time to point out meticulously issues pertaining to the content of this work. I have accordingly tried to answer the various queries and also revise the current manuscript. Please find answers to the questions raised in the lines that follow. The edited portions appear as red-colored-font in the revised manuscript.

Associate Editor

1. Minor revisions are required for typographical errors and referencing as required by reviewer 1. The authors should carefully consider the further comments of reviewer 2 to further improve the manuscript. In the absence of making the changes encouraged, the manuscript to should be adjusted to explicitly note the limitations/boundaries of the approach to date as described in the manuscript.

ANS: I thank the editor for this comment. I have corrected the typographical and also referencing errors pointed out by the first reviewer. I have also attempted to revise the manuscript to account for the comments made by the second reviewer. In this regard, I have used the example of *frugal shaft* from my earlier publication in this journal [1]. Since efforts are in progress for using this methodology in real time design, I am using the *frugal shaft* example from [1]. It should be noted that in using a table and figure from [1], I have made very minor modification to the figure (in changing N to S) and the table is listed in a different format. I have also cited the source after the captions for Figure 1 and Table 1. This example clarifies the workings of various parameters, including the one on manufacturing (topic of the

current paper) and, also ensuing calculations to compute *factor of frugality*. Other than showing the actual quantification of F^S , this simple example also describes, semi-qualitatively, resulting savings in both energy and costs.

Reviewer 1

1. I m happy with the revisions. Please fix some typographical errors (e.g., Instn Names cited in paper in lower case, Refs in garbled format etc).

ANS: I thank the reviewer for this comment. I have corrected the typographical errors related to institute names in the manuscript. Referencing has been changed to Vancouver style of RSOS thereby correcting these errors.

Reviewer 2

1. In summary, the author has tried to incorporate the changes suggested by the reviewer. The revised draft includes major changes in the way the manuscript is written, making it a much better draft. Thank you for doing that.

ANS: I thank the reviewer for this comment.

2. However, the lack of a detailed use-case highlighting relevant metrics w.r.t. a context, and no quantification of cost and energy savings makes it difficult to grasp the novelty of this approach apart from standard practices. 'Computation of frugality and validating them are critical to making a philosophical concept stronger,' - This forms the major deficiency in your paper. If the authors can discuss a use case to illustrate the computation of frugality, this paper will be highly referenced in the future.

ANS: I thank the reviewer for pointing out the relevant deficiency and do agree that a working example will lend more credence to my approach. Since getting a case-study w.r.t a context is a work in progress with its longer time-scale, I have used an existing example on the *frugal shaft* [1] from this journal.

This example, under a separate new subsection, clarifies the computation of MS parameters and also F^S . The example being a simple one brings out the working clearly without complexities associated with intricate design problems. Along with steps to computing F^S for the simple shaft, I have also added semi-qualitatively, the shaft-related savings realized in costs and energy expended. The validation for this example is that this is a readymade textbook example that lends itself to explaining clearly the workings and ramifications of the *factor of frugality*. I am in the process of applying the *frugal approach* to real time examples and hopefully will have more real time examples with validation in future publications.

Accordingly, a new sub-section (**4.1.1 Example on Factor of Frugality**) with a new table (**Table 1**); a new figure (**Figure 1**); and a new reference [2] have been added. All these changes appear in the revised manuscript as red-colored-font.

References

- [1] Rao, B. C. (2019), ‘The science underlying frugal innovations should not be frugal’, *Royal Society Open Science*, Vol. 6(5), 180421.
- [2] Ugural, A.C., *Mechanical Design of Machine Components*, CRC Press, pp. 420-423, 2015.